# Intrusion Detection Quantum Sensor Networks

**DOI:** 10.3390/s22218092

**Published:** 2022-10-22

**Authors:** Marius Nagy, Naya Nagy

**Affiliations:** 1College of Computer Engineering and Science, Prince Mohammad Bin Fahd University, Al-Khobar 31952, Saudi Arabia; 2Department of Networks and Communications, College of Computer Science and Information Technology, Imam Abdulrahman Bin Faisal University, Dammam 31441, Saudi Arabia

**Keywords:** perimeter intrusion detection, quantum sensor networks, single photon interferometry, low-power sensor, quantum sensors

## Abstract

This paper proposes a perimeter detection scheme based on the quantum physical properties of photons. Existing perimeter intrusion detection schemes, if using light, rely on the classical properties of light only. Our quantum sensor network uses the quantum property of spatial superposition of photons, meaning that a photon can simultaneously follow two different paths after going through a beam splitter. Using multiple Mach–Zehnder interferometers, an entire web of paths can be generated, such that one single photon occupies them all. If an intruder violates this web in some arbitrary point, the entire photon superposition is destroyed, the photon does not self-interfere any more and this event is detected by measurements. For one single photon, the intruder detection probability is limited theoretically but can be increased arbitrarily with the usage of a sequence of photons. We show both theoretical bounds as well as practical results of the proposed schemes. The practical results are obtained by simulation experiments on IBM Quantum platforms. The benefits of our quantum approach are: low power, invisibility to potential intruders, scalability and easy practical implementation.

## 1. Introduction

Quantum algorithms already claim a long theoretical history [1] of over two decades, with proven added capabilities over their classical counterparts. Quantum algorithms find their way in domains, such as complex data structures, search algorithms, approximation algorithms, simulations, and others [2]. Over 400 quantum algorithms are posted on the “Quantum Algorithm Zoo” so far [3]. In terms of applications, however, the most successful part of quantum computation is in the field of quantum cryptography [4], with protocols of quantum key distribution and quantum digital signatures. Quantum hardware is also stepping into existence, with small-scale quantum computers available already [5]. Experiments with quantum communications and networks is the next practical effort, which is materialized in quantum satellite communications [6], quantum network connections, and others.

The solution proposed in this paper to the perimeter intrusion detection problem belongs to the less developed area of quantum sensing. A physical intruder is observed in a physical area. The intruder is a moving object: a person, a vehicle, a device, a flying object (a drone), or any other similar entity. Conversely, the area to be protected is a physical area: for example a fenced area which surrounds sensitive or strategic objects. The area to be protected may be thought of as a fixed size, ranging from the size of a single room to the size of a set of buildings.

The element of novelty in our proposed scheme is the harnessing of genuine quantum mechanical properties, such as superposition of states, interference and collapse of a superposition following a measurement. The advantages of our quantum approach to sensing intruders, relative to their classical counterparts, are:The low power required by the photon emitter in order to generate single photons at a time.The invisibility of the detection lines, due to the same property of a single photon being present in the whole arrangement at any given time.Scalability—we rigorously define a recursive process through which an entire family of quantum sensor networks of any arbitrary size can be obtained.Easily implementable with current technology, as it relies only on beam splitters and mirrors, apart from the photon emitter.

The remainder of the paper is structured as follows. A literature review relevant to perimeter intrusion detection and quantum sensors is provided in the next section. Section 3 describes in detail the inner workings of a Mach–Zehnder interferometer, which is the central piece in our proposed family of quantum sensor networks. In Section 4, we show how multiple interferometers can be combined together in order to cover a larger surface with many different areas that can be surveilled. We also perform a detailed analysis of the probabilities of detecting intrusions in the case of one particular member of the family of quantum sensor networks. Simulation results to validate our proposed theoretical model are presented in Section 5. A summary of the most important benefits offered by our quantum approach to perimeter intrusion detection is offered in the concluding section.

## 2. Literature Review

A less developed area, at least in comparison to quantum algorithms and cryptographic applications, has emerged under the name of *quantum sensors.* In physics, quantum sensors [7] are devices that exploit specific quantum mechanical properties of quantum systems in order to measure physical quantities such as energy level, magnetic states, spin, and others. In quantum sensors, the elements measured are minute: photons, electrons, protons, small nuclei and others. As such, nano quantum sensors find applications in biology [8] and can be used as part of intelligent systems [9].

In this paper, we propose a fundamentally different understanding of the term “quantum sensor”, namely an optical arrangement that can exploit single-photon interferometry in order to detect an intruder. The building block of our scheme is the Mach–Zehnder interferometer [10], which exhibits a solid spectrum of practical implementations [11].

The quality of an intruder detection system is affected by its capability to correctly set off the alarm in the case of an intruder while keeping the false alarms, also called nuisance alarms, low [12]. Nuisance alarms can be categorized into

Alarms of unknown origin: These are alarms where the cause cannot be detected. It remains unclear whether there is a valid intruder or not.Alarms that cannot be assessed: In this case, when the alarm goes off, there is not enough information to determine the type or cause of the alarm. Mainly, it means that supporting information to determine the nature of the intruder is missing.Alarms caused by special environment conditions: These types of alarms are also false alarms, where the system goes off because of some condition in the environment. The most common cause of such events is the wind. Other conditions include heavy rainfall, floods, and earthquakes.

In terms of quality of the quantum intrusion detector developed in this paper, no practical measurements exist as yet. Nevertheless, as the system relies on photons, the system is expected to have a similar quality to optical sensors in general.

There are many commercial companies offering perimeter intrusion detection systems. In 2022, 107 companies have participated in the International Fire and Security Exhibition and Conference [13]. Depending on the product, perimeter intruder detectors are deployed on fences, underground, or wall mounted. Current detection technologies are based on video surveillance [14], optical fibers [12,15,16,17], laser scanning [18,19], and infrared beams [20,21,22].

While most video surveillance methods are passive, there are systems that are actively searching the recording for a specific target. It is the case of Zhang et al. [23], for example, where Fourier descriptors and histograms of oriented gradients are used to identify human bodies in various postures. Other approaches are based on the Internet of Things in order to implement perimeter intrusion detection for high-speed railways [24].

A comprehensive survey of optical fiber sensors in physical intrusion detection systems can be found in Allwood et al. [25]. Some of the approaches surveilled therein also use interferometry as a means for intrusion detection: an optical wave source is split using a 50–50 coupler and then directed, either through two separate fibers or in opposite directions through the same fiber, before being recombined at a detector. Any external influence on the fibers, such as, for example, the vibrations caused by an intruder’s footsteps, may cause a phase imbalance in the two waves, ruining the interference.

In contrast, our approach uses single-photon interferometry in an open air setting for light propagation, where the intrusion collapses the spatial superposition state of the photon. This leads to a low-power solution that is easy to scale and implement with beam splitters and mirrors.

## 3. Single Photon Interferometry

We begin our presentation with a detailed description of the Mach–Zehnder interferometer, which is used as a building block for the design of our quantum sensor network. The components that make up a Mach–Zehnder interferometer, as depicted in Figure 1, are two beam splitters, BS1 and BS2, and two mirrors M.

A beam splitter is an optical device typically manufactured from some transparent material but with one side that is semi-reflective. This means that when a beam of light is directed toward the beam splitter, it will let half of it pass through, and it will reflect the other half. Now, an interesting thing happens when a single photon is shot toward the beam splitter. According to quantum mechanics, the state of the photon becomes a superposition of being simultaneously reflected and passed through. This is exactly what happens with a photon hitting the first beam splitter (*BS*1) in Figure 1. It is at the same time reflected into the upper arm *U* of the interferometer, while at the same time, it continues its journey horizontally, through the transparent material of the beam splitter, into the lower arm *L*. The following matrix describes this behavior of *BS*1 precisely, from a mathematical point of view, as a quantum operator:(1)$$BS1=\frac{1}{\sqrt{2}}\left(\begin{array}{cc} e^{2i\theta } & e^{i\theta }\\ e^{i\theta } & -1 \end{array}\right).$$

This operator acts on the state space spanned by the two basis vectors:(2)$$|L\rangle =\left(\begin{array}{c} 1\\ 0 \end{array}\right)\mathrm{and}\;|U\rangle =\left(\begin{array}{c} 0\\ 1 \end{array}\right).$$
|L〉 stands for the state of a photon traversing the interferometer through the lower arm, while |U〉 represents the state of a photon going through the upper arm of the interferometer. Thus, according to this basis, a photon that approaches BS1 horizontally, from the left side, is in the initial state $|\psi_{0}\rangle =|U\rangle$. The effect of BS1 on such a photon is to change its state into a superposition of |L〉 and |U〉, corresponding to the photon simultaneously being refracted through the glass material of the beam splitter into the lower arm and being reflected by the semi-reflective surface into the upper arm. Consequently, the quantum state of the photon becomes:(3)$$|\psi_{1}\rangle =BS1|\psi_{0}\rangle =\frac{1}{\sqrt{2}}(e^{i\theta }|L\rangle -|U\rangle ).$$

The $e^{i\theta }$ phase shift in front of the |L〉 component in the equation above is due to the optical path δ covered by the photon through the transparent material of the beam splitter if refracted in the lower arm *L*. The other component, corresponding to the photon being reflected into the upper arm *U*, picks up a relative phase shift of $e^{i\pi }=-1$. A shift in phase of one half a wavelength happens every time light is incident on a surface that has a higher index of refraction than the medium the light is traveling from (air, in this case).

The two phase shifts, corresponding to |L〉, respectively |U〉, form the second column in the matrix that defines *BS*1 as a quantum operator. The other two phase shifts in this matrix, making up the first column, correspond to a superposition state of |L〉 and |U〉 created when we apply *BS*1 to a photon that hits the beam splitter from below (this scenario is not depicted in Figure 1):(4)$$BS1|L\rangle =\frac{1}{\sqrt{2}}(e^{2i\theta }|L\rangle +e^{i\theta }|U\rangle ).$$

In this case, when the photon arrives vertically, from below, to the beam splitter, the reflected component has to traverse a distance of 2δ and hence the 2θ phase shift for the lower arm base vector |L〉, while the upper arm base vector |U〉 picks up a relative phase shift of just θ, because once it reaches the semi-reflective surface, the photon goes through into the upper arm, and it is not reflected back.

The two mirrors in Figure 1 act as a single quantum operator that introduces a phase shift of $e^{i\pi }=-1$ to both the lower arm basis vector |L〉 and the upper arm basis vector |U〉:(5)$$M=\left(\begin{array}{cc} -1 & 0\\ 0 & -1 \end{array}\right)=-I.$$

The reason is again a light reflection where the index of refraction of the material the mirror is made of is higher than the index of refraction of the medium the light was traveling from (that is, air). The role of the two mirrors in the interferometer apparatus is simply to bring back together the two branches (arms), so that the two components can interfere at the second beam splitter.

Consequently, if we apply operator *M* to the quantum state $|\psi_{1}\rangle$ of a photon that enters the Mach–Zehnder interferometer horizontally, as shown in Figure 1, and is then split into a superposition by *BS*1, its quantum state evolves into:(6)$$|\psi_{2}\rangle =M|\psi_{1}\rangle =-I|\psi_{1}\rangle =\frac{1}{\sqrt{2}}(|U\rangle -e^{i\theta }|L\rangle ).$$

Finally, we can describe the effect of *BS*2 through a quantum operator with a matrix that is very similar to that of *BS*1. The only difference is that the two rows and the two columns will be swapped, due to the fact that in *BS*2, the semi-reflective surface is oriented downwards (not upwards, as in *BS*1):(7)$$BS2=\frac{1}{\sqrt{2}}\left(\begin{array}{cc} -1 & e^{i\theta }\\ e^{i\theta } & e^{2i\theta } \end{array}\right).$$

According to this operator, a photon that hits *BS*2 from below (coming from the lower arm *L*) enters a superposition state of being simultaneously reflected with a phase shift of π and refracted (passing through the beam splitter), in which case the relative phase shift acquired is θ (see the first column in the matrix above). On the other hand, when a photon arrives at *BS*2 coming from the upper arm (see the second column in the same matrix), its quantum state becomes again a superposition of |L〉 (going through) and |U〉 (being reflected and exiting vertically). As usual, each of the two components picks up a relative phase shift proportional with the optical path traversed through the glass material of the beam splitter: θ for the component that goes through and 2θ for the component that is reflected vertically.

At this point, all optical elements of the Mach–Zehnder interferometer have been precisely described in terms of quantum operators that evolve quantum states in a state space having {|L〉,|U〉} as an orthonormal basis. Now, in order to determine what happens to a photon that enters the interferometer horizontally, from the left side, we just need to apply the corresponding operators, namely BS1, *M* and BS2, in this order, on the state vector $|\psi_{0}\rangle$:
(8)$$\begin{array}{c} |\psi_{3}\rangle =BS2\cdot M\cdot BS1|\psi_{0}\rangle =BS2\cdot M|\psi_{1}\rangle \\ =BS2|\psi_{2}\rangle =\frac{1}{\sqrt{2}}\left(\begin{array}{cc} -1 & e^{i\theta }\\ e^{i\theta } & e^{2i\theta } \end{array}\right)\cdot \frac{1}{\sqrt{2}}\left(\begin{array}{c} -e^{i\theta }\\ 1 \end{array}\right)\\ =\frac{1}{2}\left(\begin{array}{c} 2e^{i\theta }\\ 0 \end{array}\right)=e^{i\theta }|L\rangle . \end{array}$$

According to Equation (Equation 8) above, the final state of the photon is not a superposition, but just the |L〉 basis vector with a global phase in front of it, which accounts for the optical path traversed by the photon through the transparent material making up a beam splitter. The physical interpretation of this is that any photon directed toward a Mach–Zehnder interferometer horizontally, from the left side, as Figure 1 shows, will always emerge horizontally, on the other side, and be detected by photon detector D1. The reason for this behavior is the *interference* taking place at the second beam splitter between the component traveling through the lower arm *L* and the component going through the upper arm *U*. This interference is destructive for the |U〉 component, canceling it from the final state, and it is constructive for |L〉, reinforcing its amplitude in the final state $|\psi_{3}\rangle$.

The interference witnessed in the experiment described above takes place as long as no attempt is made to obtain information about which of the two arms of the device the photon goes through. Such an attempt would be akin to measuring the position of the photon throughout the interferometer, and it would ruin the interference by collapsing the superposition state $|\psi_{1}\rangle$ to one of the two basis states |L〉 or |U〉, whichever is consistent with the outcome of the measurement. We show in what follows how this essential property of the Mach–Zehnder interferometer, namely, that interference can only occur if the path of the photon through the apparatus is not tempered with, can be exploited for *sensing* purposes.

## 4. Quantum Sensor

The optical elements of a Mach–Zehnder interferometer can be used for surveillance or sensing purposes. Figure 2 shows an example of such an arrangement, where the two arms of the interferometer are each monitoring an area of interest.

If nothing disturbs the path of the photon traveling through the arrangement, then photon detector D1 should register a click due to the constructive interference that reinforces the outcome that the photon emerges horizontally. However, suppose that something blocks the path followed by the photon in the upper arm of the interferometer that goes through Area 1. Such a disturbance, even if momentary, amounts to a measurement or observation of the path taken by the photon in its way from the first beam splitter to the second. Consequently, the superposition in the photon path created by BS1 will collapse to one of the two basis states with equal probability: either the photon “materializes” fully on the upper arm or it traverses the optical arrangement by following just the lower arm.

If the former situation occurs, then the photon is either absorbed or reflected away by the intruding entity, with no click registered in either D1 or D2. However, if the probabilities favor the collapse of the wave function onto the lower arm, then the photon will reach BS2 coming from the lower arm and end up in one of the two photon detectors, again with equal probability. Overall, there is a 25% chance that the photon finishes its journey in detector D1, making it impossible to detect the intrusion, and a 75% probability of a D2 click or no click at all, which indicates that something obstructed the normal intereference exhibited otherwise by the photon. The detection rate is the same if something were to disturb the lower arm of the interferometer, passing through Area 2. In the case of a simultaneous intruder event in both areas, nothing would be registered by any of the photon detectors, meaning that this type of event is always detected (with certainty) by our quantum sensor.

The main advantage of the intruder detection solution presented in Figure 2 is its efficiency in terms of the power consumption required to achieve its goal. Unlike current intruder detection arrangements that either involve a fiber optic cable as a propagation medium for light or a focused laser beam that travels through open air, our solution employs the quantum effects of an individual particle such that at any moment, there is at most one photon in the entire arrangement (interferometer). In other words, the photon emitter in our scheme has to be tuned down to levels low enough in order for a photon to escape the emitter every few milliseconds or any other time interval that is considered small enough for an intruder not to be able to pass by undetected.

A second advantage also stemming from the fact that only a single photon travels through the quantum sensor at any given time is the fact that the detection lines (optical paths of the interferometer) are not visible to a potential intruder in the visible spectrum or otherwise. This stands again in contrast to the other detection methods, based on a laser beam traveling through air or a fiber optic cable.

Furthermore, the arrangement depicted in Figure 2 can be extended in order to cover a larger area in the following way. After the initial beam splitter creates the superposition in the photon position, each of the two paths, the lower arm and the upper arm, goes again through a full Mach–Zehnder interferometer before reuniting at the final beam splitter. The entire arrangement together with the possible areas that can be surveilled in this way are shown in Figure 3.

Because of the same interferometric effect described in detail in the previous section, if nothing disturbs the multiple paths taken in quantum superposition by the photon and the two arms created by a beam splitter are always of equal length, then the photon will always emerge horizontally from the apparatus and hit detector D4. If any of the other photon detectors is triggered or none of the detectors registers a click, then something prevented the normal interference from taking place, which is an event assimilated with an intrusion. Which particular detectors are clicking and what is the probability of detecting an intrusion depends entirely on the particular area(s) where the intrusion takes place. In the following, we analyze in detail the consequences of an intrusion in each of the areas depicted in Figure 3.

**Case I**: Intruder disrupts the optical path traversing Area 1 and/or Area 2.This scenario amounts to a quantum measurement on the position of the photon after the initial beam splitter. The superposition may collapse onto the upper arm, with 50% chance, or onto the lower arm, again with 50% chance.  Upper Arm: The photon is absorbed or reflected by the intruder, and none of the detectors register the photon. The intruder has been detected.  Lower Arm: The photon traverses Area 3 and the entire Mach–Zehnder interferometer that follows, reaching the final beam splitter coming from Area 14. Here, there is a 50–50 chance of being detected by either D3 or D4. If D3 clicks, then the intruder is correctly detected. If D4 clicks, the outcome is inconclusive.Overall, considering both options, there is a $\frac{3}{4}=75\%$ probability of detecting the intrusion (no click or D3 clicks) and a $\frac{1}{4}=25\%$ probability of an inconclusive outcome (D4 registers a click).**Case II**: Intruder obstructs the photon path going through Area 3. The probabilities in this case are the mirror of Case 1. A collapse to the lower arm will result in no click. In case the superposition collapses to the upper arm, the photon reaches the final beam splitter coming from Area 12, and it will still be detected with equal probability by D3 or D4. Hence, the overall probability to successfully detect the intruder is again $\frac{3}{4}=75\%$.**Case III**: Intruder blocks photon path in Area 4 and/or Area 6. This case corresponds to a more complex measurement of the photon position in which the quantum state being measured is an equiprobable superposition of the photon going through A6, A7, A8 and A9 at the same time:
(9)$$|\Psi \rangle =\frac{1}{2}(|A6\rangle +|A7\rangle +|A8\rangle +\left|A9\rangle \right)$$This superposition can collapse on either of the four states with equal probability, namely $\frac{1}{4}$.  Collapse on A6, equivalent to A4: The photon is absorbed/reflected by the intruder and the apparatus does not click. The intruder is detected with a chance of the collapse $\frac{1}{4}$.  Collapse on A7: As the photon continues through the apparatus, it has an equal probability to choose horizontal or vertical paths on each encountered beam splitter. Thus, D1 will register the photon with overall probability $\frac{1}{4}*\frac{1}{2}=\frac{1}{8}$, and D3 and D4 exhibit the same probability, each of $\frac{1}{4}*\frac{1}{4}=\frac{1}{16}$. The intruder is detected by D1 or D3 with a combined probability of $\frac{1}{8}+\frac{1}{16}=\frac{3}{16}$, and the measurement is inconclusive for D4, $\frac{3}{16}$.  Collapse on A8 or A9: Since the measurement describing this case cannot distinguish between the two regions, the interference taking place in the beam splitter just before A13 is undisturbed. Coming from A14, the photon will end up either in D3 or D4, giving a probability of $\frac{1}{2}*\frac{1}{2}=\frac{1}{4}$ positive detection probability and the same $\frac{1}{2}*\frac{1}{2}=\frac{1}{4}$ inconclusive measurement probability. Overall, there is a $\frac{1}{16}+\frac{1}{4}=31.25\%$ chance of an inconclusive outcome and a 68.75% probability of detection.**Case IV**: Intruder blocks photon path in Area 7 and/or Area 10.This case is very similar to the previous one, with the difference that the intruder absorbs or reflects the photon when the superposition state collapses on A7, not A6. Otherwise, the overall probability of detection is the same, namely 68.75%.**Case V**: Intruder disturbs the optical path going through Area 5 and/or Area 8.This scenario corresponds to a quantum observation of the path taken by the photon through the extended quantum sensor with three possible outcomes: the photon traverses the upper interferometer, exiting the beam splitter before A12 horizontally and arriving at D3 or D4; the photon is absorbed/reflected by the intruder while going through A5 and/or A8 (no click); the photon goes through A9 and A11, ending up in D2 (with 50% chance) or in D3 (with 25% probability) or in D4 (again with 25% probability). Overall, there is again a 31.25% chance of an inconclusive outcome (detector D4 registers a click) and a 68.75% probability of detection (when there is a click in D2, D3 or no click at all).**Case VI**: Intruder disturbs the optical path traversing Area 9 and/or Area 11.The quantum measurement assimilated with this scenario is very similar to the one in Case V, except when the photon goes through A5–A8, it is going to be detected by D2, D3 or D4, and when it traverses A9–A11, it is going to be absorbed or reflected. Therefore, the overall probability of detection is identical to that of Case V (68.75%).**Case VII**: Intruder obstructs the path of the photon in Area 12.This amounts to a quantum observation that determines on which interferometer the photon really did go through: the one in the upper arm or the one in the lower arm. If it is the former, then the photon is absorbed/reflected by the intruder in Area 12 with no click registered in any of the detectors. In the case of the latter, the photon exits the Mach–Zehnder interferometer in the lower arm by going through Area 13, continues onward through A14 and finishes its journey in D3 or D4. Consequently, we have a 75% probability of detecting the intrusion in this scenario (no click or D3 registers a click) and a 25% probability of an inconclusive outcome (D4 registers a click).**Case VIII**: Intruder blocks the path of the photon in Area 13 and/or Area 14.This last case corresponds to the same quantum observation as in the previous case but with the two possible outcomes switched. The photon is registered by D3 or D4 if it goes through the upper interferometer, and it is absorbed/reflected when it traverses the lower interferometer. This change does not influence the overall probability of detecting the intruder, which remains at 75%.

Note that each of the cases surveyed above corresponds to a disturbance in the optical path between two consecutive beam splitters (e.g., Case IV covers the lower arm of the upper interferometer). Simultaneous intruder events (two or more cases that are active at the same time) are possible and may result in a higher probability of detecting the intrusion, depending on the particular areas that are being disturbed. The ability to guard a large total area against potential intruders with a single photon at a time is a remarkable property of the extended quantum sensor from Figure 3. The downside, however, is that the arrangement does not offer much information about where exactly the intrusion has occurred, as classical intrusion detection systems are able to do. Only detectors D1 and D2 are able to offer some clue with respect to the whereabouts of the intruder. A click registered in D1 means a disturbance somewhere in the upper interferometer (Areas 4, 6, 7 or 10). Similarly, a click registered by D2 points to an intruder obstructing one of the arms in the lower interferometer (Areas 5, 8, 9 or 11). Otherwise, a click registered by D3 or D4 or the absence of any click may result from an intrusion in any of the areas depicted in Figure 3.

The analysis above shows that an intruder can be detected using one photon only with a probability of 75% or 68.75% depending on the position. If this were the entire capability of the system, it would be a modest success. As with quantum protocols in general, the detection rate can be made arbitrarily high by increasing the number of photons. This can be performed directly, as a photon emitter generates photons at a rate that can be set by the user. For *n* photons, denote the detection probability for each photon to be $p_{i}$, where 1≤i≤n; then, the detection probability of all photons together is $1-\sum_{i=1}^{n}\left(1-p_{i}\right)$, which approaches 1 as *n* grows.

In practice, the rate of firing photons can be set based on the formula
(10)$$f=\frac{\Delta t}{n},$$
where Δt represents the time interval of the shortest intruder event considered (the time between the moment the optical path of the photon is interrupted and the moment it is restored) and *n* is the minimum number of photons that achieve a satisfactory detection rate. In this way, for every possible intrusion, there is a sequence of *n* photons, and each has a chance of detecting the intrusion. Furthermore, for a sequence of *n* photons, not only is the overall detection rate improved but the intruder location information is improved as well, as the probabilities of D1 and D2 to register a click will also increase.

The extended quantum sensor that we have defined and analyzed above is just a member of a family of quantum sensor networks that can be defined recursively as follows. A single Mach–Zehnder interferometer is a member of the family (the base case). Two members of the family can be joined together to form a new member by placing one in the upper arm and the other in the lower arm of a Mach–Zehnder interferometer. Repeating this process can lead to a quantum sensor network that can surveil, in principle, an arbitrary number of areas, still with just one photon in the entire arrangement at the time. As an example, Figure 4 shows how two extended quantum sensors can be combined together in order to create a new member of the family.

Naturally, in practice, there are limitations with respect to how far this recursive process can be pushed and, consequently, how extensive the resulting quantum sensor network can be. This is due to the practical limitations of the optical elements involved in the arrangement, the mirrors and the beam splitters, which may not always behave in the ideal way described at the beginning of the paper. For example, the more devices a photon has to go through, the higher the probability that it will eventually be absorbed, leading to a false positive intruder alert. However, even with a single Mach–Zehnder interferometer, the area that can be surveilled may be quite large if the distance between the optical elements is large.

Another observation regarding the topology of the members of the recursive family of quantum sensor networks is that all three examples pictured above, the basic quantum sensor (Figure 2), the extended quantum sensor (Figure 3) and the quantum sensor network of Figure 4, have a balanced or symmetric topology, in the sense that the upper arm of the network is identical with the lower arm. According to the recursive rule used to construct a new member, this need not be the case. It is, for example, possible to combine an extended quantum sensor with a basic quantum sensor as long as the length of the optical path through the two components is the same, so that the two branches can interfere at the end. However, a symmetric topology may be preferred in most cases, as it offers the same degree of detail in surveilling the space covered by the two branches.

Although all the arrangements presented above may be labeled as planar, in the sense that all components lie in the same plane, it is important to emphasize that quantum sensor networks with a 3D topology are also possible. By introducing additional mirrors and adjusting their orientation, we can create intricate optical paths that can cover any three-dimensional space that needs to be protected from potential intruders. Again, the only requirement that needs to be satisfied is that the various branches of the interferometers composing the network have exactly the same length, so that interference can take place.

## 5. Simulations

The performance of our proposed quantum sensor network will depend, in practice, on the quality of the optical elements used and other details pertaining to the actual implementation on the ground. However, as a proof of concept for our method, we have simulated the quantum superposition state of a photon traversing the optical arrangement in the extended quantum sensor (Figure 3) on the IBM Quantum platform.

From the comprehensive analysis performed in the previous section, we have selected Case I and III to simulate, since they are two of the most representative ones and the others yield similar measurement probabilities. The IBM Quantum platform offers both a classical simulator that can mimic quantum states and operations on them (quantum gates) as well as a set of small-scale actual quantum machines capable of manipulating a handful of qubits or so. For our quantum simulations, we have chosen the *ibmq_belem*,which is a quantum computer with five qubits.

In order to simulate Case I, we have used two qubits: one to model the function of the first beam splitter, which creates the upper arm (Area 1) and the lower arm (Area 3) of the network, and a second one to represent the very last beam splitter, whose exits end up in detectors D3 and D4. The comparative results of the simulation are presented in Figure 5, where each bar represents the average value of 1024 trials. The first bar shows the results obtained on a classical simulation of quantum behavior. The second bar shows the measurements performed on the IBM—belem quantum computer, and the third bar represents the theoretically expected values. We can see that both the classical and quantum simulations are very close to the results of the theoretical analysis of Case I: 50% probability that the photon is absorbed/reflected by the intruder (no clicks), and 25% probability that the photon triggers detector D3, respectively D4.

In the simulation of Case III, two additional qubits are necessary in order to model the two beam splitters making up the upper Mach–Zehnder interferometer. The results of the simulations are depicted in Figure 6. Although the values obtained are again close to the expected probabilities from the theoretical analysis, we note that the quantum simulations deviate more from them compared with the classical simulations. This is certainly to be expected, since the values on the quantum simulations are obtained on an actual physical quantum machine whose operation is subjected to quantum errors.

Overall, these simulations are an indication that the quantum method for intrusion detection, described theoretically in this paper, is feasible and a practical implementation is warranted.

## 6. Conclusions

In this paper, we have proposed a detection system that applies the principles of single photon interferometry to the problem at hand. One of its main advantages is that it requires a very low-powered photon emitter, since the proposed scheme requires only one photon to be present in the arrangement at any given time. The solution is easily scalable to any desired complexity due to a straightforward recursive process of combining smaller networks into larger ones, which is a process limited only by practical considerations. In addition, the light generated by the photon emitter can be in any range, including light from the visible spectrum, since the low intensity (one photon at a time) makes it invisible anyway.

Finally, we note the ease of implementation for any member of our family of quantum sensor networks, requiring only beam splitters and mirrors, besides the photon emitter. Moreover, the quantum mechanical principles underlying our design of quantum sensor networks can be applied to any particle that can exhibit self-interference, such as, for example, electrons. Naturally, in that case, the Mach–Zehnder interferometer with its optical elements will have to be replaced with an analogous experimental setup that allows an electron to interfere with itself [26]. Although practical implementations are beyond the scope of this paper, classical and quantum simulations performed on the IBM Quantum platform validate the theoretical model and leave the door open for future physical realizations of our proposed system.

## Figures and Tables

**Figure 1 sensors-22-08092-f001:**
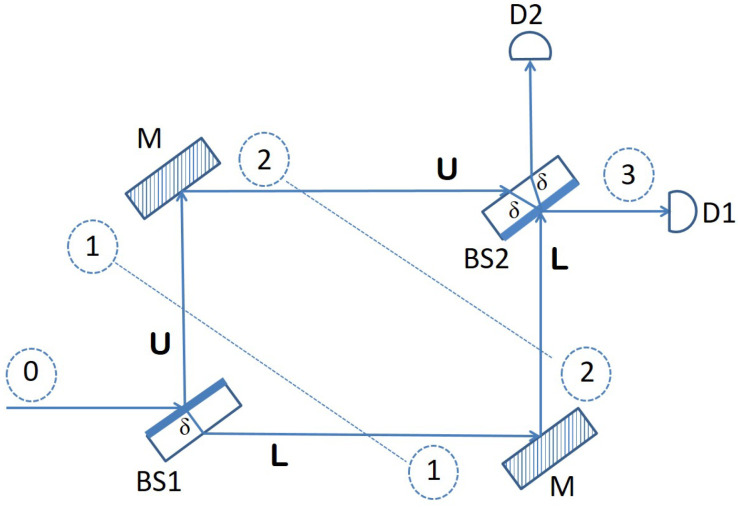
Mach–Zehnder interferometer.

**Figure 2 sensors-22-08092-f002:**
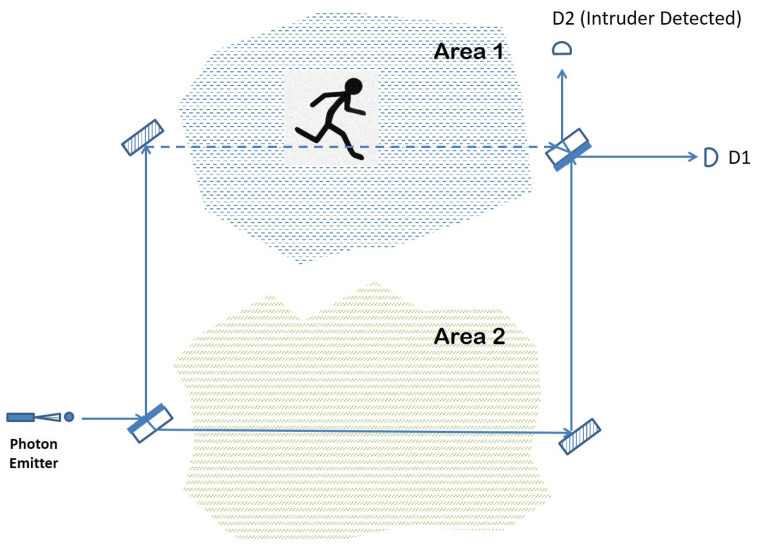
Quantum sensor based on single-photon interferometry.

**Figure 3 sensors-22-08092-f003:**
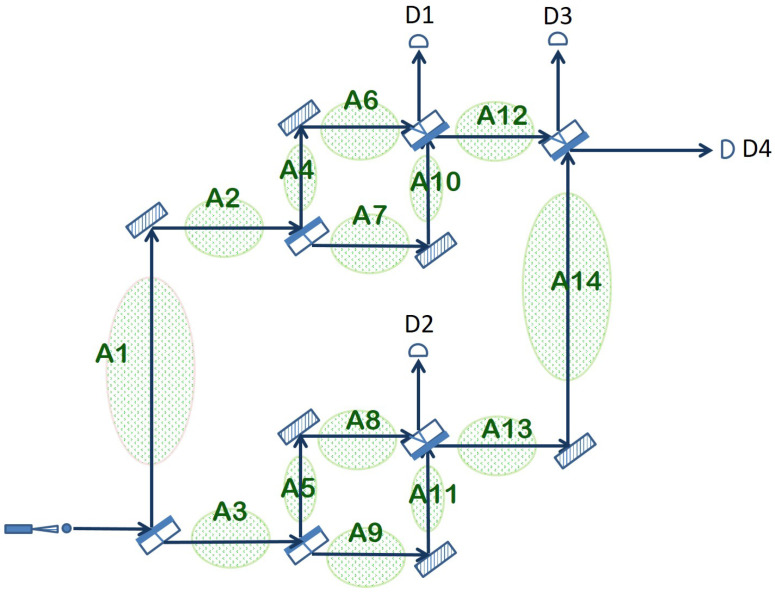
Extended quantum sensor obtained by placing an additional Mach–Zehnder interferometer in each of the two arms of the original interferometer.

**Figure 4 sensors-22-08092-f004:**
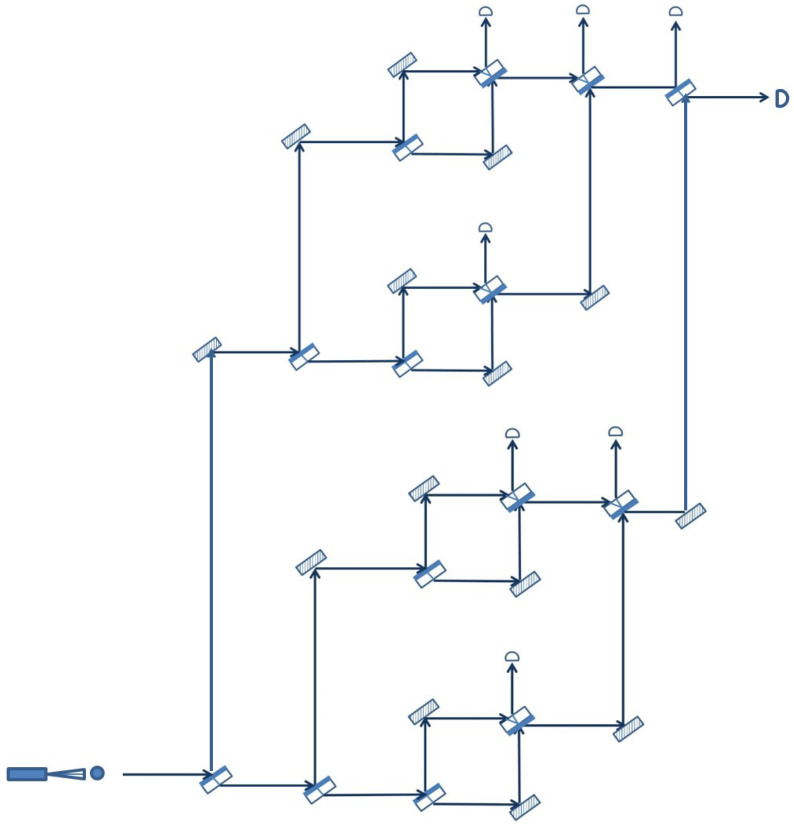
Quantum sensor network obtained recursively from two extended quantum sensors.

**Figure 5 sensors-22-08092-f005:**
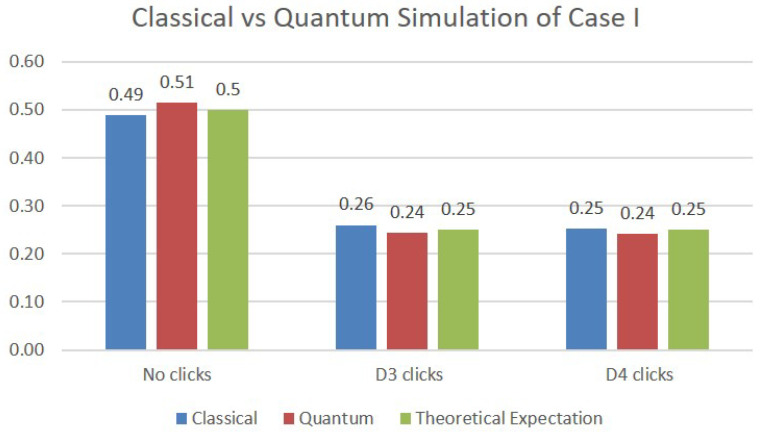
Simulations results for Case I.

**Figure 6 sensors-22-08092-f006:**
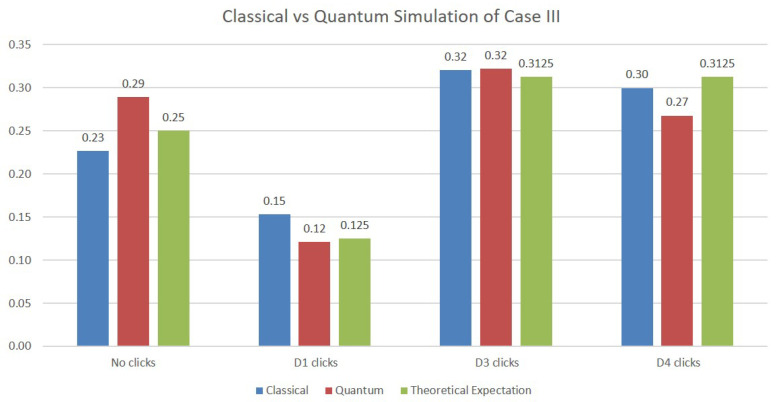
Simulations results for Case III.

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
