# Peer review of "Intrusion Detection Quantum Sensor Networks"

_sensors, 2022, doi:10.3390/s22218092_

Round 1
Reviewer 1 Report
ID: sensors-1961176
Title: Intrusion Detection Quantum Sensor Networks
This paper proposes a new scheme to detect intruders in quantum sensor networks. The topic sounds good and best fits the scope of the journal. However, the major concern is related to the experimental demonstration of the intruder detection system and performance comparison with other existing methods. How can authors measure the quality and performance of an intruder detection system for quantum sensor networks? There are no experimental and simulation results given as evidence to measure the quality and suitability of the intrusion detection system. Need simulation or experimental demonstration to validate the quality and performance of the proposed scheme as compared to classical approaches. However, the authors should also address a few minor points as well as given below:
1) The Abstract is not very enlightening, be more insightful about the proposed scheme and findings.
2) To highlight the existing research and debates relevant to perimeter intrusion detection systems, optical systems, and more specifically intrusion detection systems in quantum sensor networks. The authors should add a new section for the literature review after the introduction section. And also, should highlight the research gap at the end of the literature review.
3) The conclusion part is somewhat too long and the given points are not really justified in detail based on the evidence presented.
4) The paper needs very careful proofreading as some typos and grammatical errors exist.

Author Response
We thank the reviewer for the constructive comments formulated. We feel that they have improved the quality of the presentation.
In response to the main concern of the reviewer, we have added a new section (Section 5 "Simulations") that presents the results of simulating the quantum state of a photon going through the sensor network and the measurements assimilated with an intrusion event on the IBM Quantum platform. An actual implementation of a quantum sensor network on the ground, with real photon emitters, mirrors and detectors is outside of the scope of this work and beyond our resources. Both the classical and quantum simulations show that the system should behave close to the theoretical predictions derived in the analysis from Section 4.
The other minor points have been addressed as follows:
1) The Abstract has been reformulated to better explain the approach used and
results obtained.
2) A new section "Literature Review" has been added after Introduction, with the
relevant context of perimeter intrusion detection (the total number of references has been expanded to 26). We explicitly mention at the end of the section the gap that we address in our paper: using quantum properties
of light for intrusion detection, something that none of the previous methods is
doing.
3) The Conclusion section has been shortened, as suggested.
4) The entire paper has been proofread again and a couple of typos have been fixed. If the reviewer still considers that some sentences contain grammar errors, we kindly ask him to point them out, so that we can correct them.
Reviewer 2 Report
The proposed approach is interesting as a novelty, however I have some concerns with the manuscript:
- The main justification for the proposed approach is that its power consumption is portrayed to be lower than classical solutions. However, in practice, many photon emitters are tonned-down lasers. Furthermore, the installation cost might well overshadow those of other alarm systems. It would be interesting to provide a comparative analysis in terms of the cost of the networks.
- The authors state that their system is 'invisible' yet there seems to be a large number of mirrors which can be readily perceived.
- It is also asumed that the interferometers would be 'easy to implement in practice.' However, beyond the observation of the required reliability of materials, real scenarios come with all kind of imperfections which might intefere with a sensitive system. For starters, spaces tend to not be square or offer clear lines of sight.
- Could the proposed approach be applied to three-dimentional spaces? The most usual intrussion detectors can monitor a space rather than a line of sight.
Author Response
The concerns raised by the reviewer are legitimate and we have addressed them as best we could as follows:
- in terms of power consumption, a photon emitter that only shoots one photon at a time achieves the minimum possible power. Most classical solutions to intrusion detection that use some form of light work with beams or light waves (trains of photons) rather than individual photons. Therefore, we stand by the claim that our setting is more energy efficient than the vast majority (if not all) of the existing solutions for perimeter intrusion detection.
- we agree that it would be interesting to evaluate an actual implementation of our quantum sensor network on the ground, with real photon emitters, mirrors and detectors, but this is outside the scope of this work and beyond our resources.
However, we have added a new section (Section 5 "Simulations") that presents the results of simulating the quantum state of a photon going through the sensor network and the measurements assimilated with an intrusion event on the IBM Quantum platform. Both the classical and quantum simulations show that the system should behave close to the theoretical predictions derived in the analysis from Section 4.
- The Abstract has been reformulated to better explain the approach used and
results obtained. Also, the literature review has been expanded into a separate
section, in order to better place our work into context.
- it is true that the actual optical devices (the photon emitter, beam splitters,
mirrors, detectors) are not invisible and this is true in general, for the equipment of all perimeter intrusion detection systems. The "invisibility" refers only to the line that sounds the alarm when crossed (the "detection lines" or "optical paths", as we explain it in the paper at the end of page 6). In our system, the photons generated can be of any wavelength, since individual photons cannot be perceived with the naked eye, but in classical solutions based on a laser beam, it is imperative the light used is outside the visible spectrum in order to make it invisible.
- we have added a new paragraph at the end of Section 4 in which we explicitly
mention that mirrors can be oriented such that any 3D space can be monitored for possible intrusions.
Round 2
Reviewer 1 Report
I think most of my comments have been addressed with either new analysis or necessary discussions. The revised version of the manuscript appears to be good. Therefore, I have no further comments.